# Study on the Control Method of Sidewall Taper in Electrolytic Broaching of Micro Multi-Grooves

**DOI:** 10.3390/mi13122062

**Published:** 2022-11-24

**Authors:** Jia Liu, Hao Wu, Feng Gao

**Affiliations:** College of Mechanical and Electrical Engineering, Nanjing University of Aeronautics & Astronautics, 29 Yu Dao Jie Street, Nanjing 210016, China

**Keywords:** electrochemical broaching, forming process simulation, cathode tool design, golden section method, experimental verification

## Abstract

Micro multi-grooves are important functional structures widely used in new heat exchanger types, chemical reactors, and other applications. Electrolytic broaching is an efficient and low-cost technology for processing micro multi-grooves. In the conventional electrolytic broaching of multi-grooves, the cathode tools are usually designed as a wedge-shaped tooth structure array with a constant tooth width, and the sidewalls are covered with insulating layers. The machined groove sidewall is always tapered because of stray current corrosion, which strongly affects the groove contour accuracy. Cathode tools with variable tooth width structures are proposed to solve this problem. Based on the simulation results of the electrolytic broaching anode forming process, the optimal front tooth width is obtained through the golden section optimization method, and comparative tests of the conventional and optimized cathode tools were carried out. At an electrochemical broaching feed rate of 120 mm/min, array microgrooves with widths of about 550 μm and depths of about 520 μm were processed. With the optimized variable tooth width tool, the sidewall tapers of the grooves were reduced from 7.254° to 0.268°. The experimental results verify the effectiveness of the simulation and cathode structure optimization.

## 1. Introduction

Multi-grooves are typical functional structures for increasing the heat exchange and reaction area and have been widely used in heat exchangers, chemical reactors, and other applications. With the rapid development of modern technology, the widespread use of microstructures, such as bipolar plate microgrooves, provides better mass and heat transfer performance [1,2,3,4,5,6,7,8,9,10,11,12,13]. New heat exchanger types and chemical reactors have been designed with multi-grooves into the submillimeter scale. Because of the large surface area (usually hundreds or even thousands of square centimeters) of micro multi-grooves on parts, severe tool wear may occur during their milling. The problems of efficiency and cost in processing micro multi-grooves over large areas have strongly affected their broad application to new products.

Micro-milling is a common processing technology for micro multi-groove manufacture. Rodríguez et al. [14] used a coolant in micro-milling, which improved micro-channel-achieved dimensions and surface finish. They also found that micro-channels in aluminum have better quality than in copper. Alwin et al. [15] focused on developing an analytical model to predict the process geometry parameters by adjusting tool runouts and different tool paths in micro-milling to avoid tool breakage. Wang et al. [16] developed a titanium-based cermet micro-milling cutter, demonstrating that the surface quality of microgrooves milled by a cermet micro-end milling cutter was higher than those milled by a coated Tungsten carbide micro-end milling cutter. However, tool wear and burr problems in machining micro multi-grooves over large surface areas still hinder the improvement of efficiency and the reduction in costs.

Micro electrical discharge machining (MEDM) is a potential machining technology for micro multi-groove manufacturing. Hung et al. [17] explored the feasibility of manufacturing micro metal bipolar plates using electric discharge technology and successfully processed 500-μm-width grooves over a 20 mm × 20 mm area of SUS316L material. Tsai et al. [18] introduced a smart auxiliary tool vibration device that could improve the efficiency of deep groove EDM. They compared the material removal rate, tool electrode wear rate, and surface roughness. The experimental results showed that the material removal rate of vibration-assisted EDM was better than that of the copper electrode and copper–tungsten electrode EDM. However, MEDM technology also has machining efficiency and tool wear challenges in large-area micro multi-groove processing.

Chemical corrosion is another potential low-cost manufacturing technology for micro multi-grooves. Chemical etching technology plays an essential role in producing heat transfer channels for printed circuit heat exchangers from the micron level to the millimeter level. Xin et al. [19] investigated the influence of etching the temperature and spray pressure on the etching rate, lateral erosion, and surface roughness. The optimal combination of the etching parameters was obtained, which is significant for realizing a low-cost, high-efficiency, and controllable manufacturing process. Although the chemical etching method has cost advantages in processing large-area micro multi-grooves, it can only process circular arc grooves, and it is difficult to produce consistently accurate microgrooves.

Electrochemical machining (ECM) is a non-contact machining technology based on the principle of the electrochemical reaction of anodic dissolution to remove materials. It has many advantages, such as no tool loss, high machining efficiency, and good surface quality and machinability independent of the material mechanical properties [20]. Compared with other machining technologies, ECM has significant efficiency and cost advantages in micro multi-groove machining. In the ECM of micro multi-grooves, there are three main processing modes: copy mode, jet mode, and broaching mode. The advantage of the copy mode is that the whole surface group groove structure can be machined by simple one-way feeding, and the geometric contour accuracy of the groove is easily controlled. The copy mode is mainly used in the ECM of micro multi-grooves over medium and small areas.

Zhao et al. [21] conducted a coupled fluid–solid simulation in which the designed hollow sheet cathode with reinforcing ribs improved the stability of the flow field when machining a deep and narrow groove structure. Liu et al. [22] designed a multifunctional cathode, proposed a new constrained flow mode and conducted experiments on the pulse electrochemical machining of metal bipolar array grooves. The results showed that multiple independent machining areas could effectively improve the speed uniformity of each channel. Jiang et al. [23] proposed a variable cross-section channel contraction cathode structure. The flow rate of the electrolyte in the electrode gap gradually increased to better remove the electrolytic products. The experimental results showed that the method effectively improves the feed speed and ensures groove depth uniformity.

Jet mode is a machining method that uses the electrochemical corrosion effect of the energized electrolyte jet to remove the workpiece material. The formation of the group groove structure is realized through the trajectory movement of the jet nozzle. Arc-shaped and cylindrical V-shaped microgrooves fabricated on stainless steel by jet electrochemical machining by Natsu et al. [24]. Clare et al. [25] adjusted the spray angle to 22.5° relative to the sweep direction to produce a good-quality machined microgroove structure. The continuous free jet electrochemical machining process was simulated by Oschatzchen et al. [26] using the finite element method. Because the groove depth of the single jet machining is relatively shallow, the flow fields between two adjacent grooves affect each other. Therefore, the jet mode is unsuitable for manufacturing deep and densely arranged micro multi-grooves.

Ultraviolet laser processing is also an important technology for machining microgrooves. The laser processing and manufacturing capability can reach the level of a micrometer. Tang Yu et al. [27] developed a simple laser irradiation method to fabricate the microscale titanium dioxide structure and observed its morphology through a scanning electron microscope and X-ray photoelectron spectroscopy. Through the analysis of high-speed photography, it was found that these micro-grooves played an important role in enhancing water transmission performance, which provides inspiration for the application integration of such promising functional surfaces.

The broaching mode is an efficient method of machining through straight grooves because of its high efficiency in machining spiral grooves on the inner wall of gun barrel rifling [28]. Tang [29] successfully machined a mixed rifle with a large winding angle and deep grooves using electrolytic broaching. He developed the gun tube blank and cathode working teeth model, wrote the computer program, carried out simulation processing, and shortened the cathode development cycle.

In electrolytic broaching (ECB) micro multi-grooves, the side of the wedge-shaped tooth tools must be covered by an insulating layer to control the microgroove width, and the front and rear tooth widths are usually designed to be equal. During processing, wedge-shaped tooth tools gradually move into the workpiece with the horizontal feeding of cathode tools. The upper part of the microgroove is corroded by a stray current, resulting in the sidewall taper and strongly affecting the groove contour accuracy. Cathode tools with variable tooth width structures are proposed to solve this problem. By gradually expanding the tooth width from front to rear, the sidewall taper of the microgroove is corrected. Simulations of the anode profile dynamic forming process and cathode tool parameter optimization of ECB micro multi-grooves were conducted to determine the optimal cathode tool structures. Then, comparative tests of conventional and optimized cathode tools were carried out to evaluate the effectiveness of the simulation and optimization results.

## 2. Principle of ECB Micro Multi-Grooves

A schematic diagram of the conventional ECB method for micro multi-grooves is shown in Figure 1. The principle of ECB method is shown in Figure 1a. The wedge-shaped tooth cathode tools and the workpiece are connected to the negative and positive poles of the power supply, respectively. The front and rear widths of each wedge-shaped tooth on the cathode tools are equal. Only the lower end face of each wedge-shaped tooth is a bare metal surface, and the other surfaces are covered with insulation layers. Insulating sheets are installed between each adjacent wedge-shaped tooth to expose the wedge-shaped teeth at a certain height. In electrolytic broaching, the insulating sheets’ lower end faces contact with the workpiece’s upper surface, and the electrolyte flows into the machining areas with wedge-shaped teeth from the gaps between the insulating sheets. With the cathode tool moving horizontally along the front of the wedge-shaped teeth, the rear wedge-shaped teeth surfaces protruding from the insulating sheet gradually feed into the workpiece surface and machining micro multi-grooves after the wedge-shaped teeth of the cathode tools pass.

The forming process at different stages when the wedge-shaped tooth tool passes through sections of the workpiece is shown in Figure 1b. Because the front part of the wedge-shaped teeth shrinks in the insulating sheet, there is an initial machining gap between the bottom surface of the wedge-shaped teeth and the workpiece surface at the initial machining stage. With the continuous feed of the wedge-shaped tooth tool, the wedge-shaped teeth gradually protrude out of the insulating sheet and gradually enter the workpiece surface. The end of the wedge-shaped tooth completely enters the inside of the workpiece until the final stage of processing passes through the section and processes the micro multi-grooves. Although the sidewalls of the wedge-shaped teeth are insulated, the stray current at the edge of the machining gap on the bottom surface continues to cause secondary corrosion on the machined surface. Therefore, the pronounced sidewall taper is produced by traditional cathode tools with equal front and rear widths, severely affecting the contour accuracy of the micro multi-grooves.

Aiming to address the problem of the sidewall taper in the ECB of micro multi-grooves, cathode tools with variable tooth width structures are proposed, with the width of the wedge-shaped tooth tool gradually increasing from front to rear. The assembly body of proposed ECB method is shown in Figure 2a. The gradual expansion of the tooth width compensates for the sidewall taper caused by secondary corrosion to improve the contour accuracy of the sidewall of the micro multi-grooves.

The forming process at different stages is shown in Figure 2b. However, because the sidewalls of the wedge-shaped teeth are insulated, a reverse sidewall taper occurs if the width of the wedge-shaped teeth increases too fast, even causing interference between the tool and workpiece. Therefore, the wedge-shaped teeth’s structural parameters must be optimized to control the micro multi-grooves’ sidewall taper.

## 3. Simulation of ECB Forming Process and Tool Structure Optimization

### 3.1. Forming Process Simulation Model and Boundary Conditions

In the ECB process, the feed rate of the wedge-shaped teeth is relative to the workpiece and is determined by the partial speed of the cathode tool feed rate in the direction perpendicular to the anode surface. The relative feed speed is directly related to the structural parameters of the wedge-shaped teeth. The structure of wedge-shaped tooth is shown in Figure 3a. The vertical and horizontal sections of a single variable-width wedge-shaped tooth are shown in Figure 3b,c, which is section B and section A, respectively. When the variable-width ECB cathode tool is fed at the rate $\nu_{c}$, the feed rate between the cathode tool tooth and the workpiece anode along the slot depth direction is calculated from Equations (1) and (2). The feed rate between the cathode tool tooth and the workpiece anode along the slot width direction is calculated from Equations (3) and (4).
(1)$$v_{c}\sin\alpha =v_{n}$$
(2)$$\tan\alpha =\frac{H_{2}-H_{1}}{L}$$
(3)$$v_{c}\sin\beta =v_{b}$$
(4)$$\tan\beta =\frac{W_{2}-W_{1}}{2L}$$

In Equations (1)–(4), *α* is the inclination angle between the cathode tooth and horizontal plane, $H_{1\;}$ is the front tooth height, $H_{2\;}$ is the rear tooth height, *L* is the wedge tooth length, *β* is the angle between the cathode tooth and the vertical plane, $W_{1}$ is the front tooth width, and $W_{2}$ is the rear tooth width. $\nu_{n}$ is the feed rate between the cathode tooth and workpiece anode along the groove depth direction, and $\nu_{b}$ is the feed rate between the cathode tooth and workpiece anode along the groove width direction.

Based on previous research, the ECB cathode tool structural parameters are shown in Table 1. The forming process at different stages, when the wedge-shaped tooth tool passes through a particular section of the workpiece, is shown in Figure 1b. In ECB processing, the electrolyte flows perpendicular to the cross-section of the simulation model, and the electrolytic products, such as bubbles, anodic dissolution products, and Joule heat, are approximately uniform over the cross-section. In addition, because the wedge tooth length is only 20 mm, there is no apparent accumulation of the electrolytic products over the short electrolyte flow length. Therefore, the influence of electrolytic products on the anode-forming process is ignored in the simulation.

The electric field distribution plays a crucial role in the final groove contour forming size. Therefore, the current field and deformation geometry modules were selected to carry out the electric field simulation in COMSOL using the transient simulation module. This paper simplifies the three-dimensional model of the machining gap into a two-dimensional longitudinal section model to reduce the required calculation time. The dynamic forming simulation model of a single wedge tooth passing through the workpiece section is shown in Figure 4. According to electric field theory, the electric potential distribution in this process is dominated by the electric field, which is described by the Laplace equation:(5)$$\nabla^{2}\varphi =\frac{\partial^{2}\varphi }{\partial^{2}x}+\frac{\partial^{2}\varphi }{\partial^{2}y}=0$$
with boundary conditions:(6)$$\varphi_{1}=0\;(\mathrm{Cathode}\;\mathrm{side})$$
(7)$$\varphi_{3}=20\;V\;(\mathrm{Anode}\;\mathrm{side})$$
(8)$$\frac{\partial \varphi }{\partial n}=0\;(\mathrm{Insulation})$$

Line 1 in Figure 4 is the low-end face of the cathode tooth, moving vertically and straight downward at speed $\nu_{n}$. Line 2 is the sidewall of the cathode tooth, which moves horizontally along a straight line at speed $\nu_{b}$ to the left and right sides. Under the electrochemical anodic dissolution reaction, the workpiece material is dissolved into the electrolyte at the rate $\nu_{3}$ which obeys Faraday’s law:(9)$$v_{3}=\eta \omega i$$
where *η*, *ω*, and *i* represent the electrolytic processing efficiency, the electrochemical equivalent of the anode material volume, and the current density on the workpiece surface, respectively.

During the simulation of the machining cycle, the entire pulse-on time is divided into several small-time intervals Δ*t*. At each Δ*t*, the electric field distribution in the machining gap is first calculated using Equation (5). Then, the slight displacement Δ*d* along the normal direction of the anode surface is calculated using Equations (9) and (10).
(10)$$\Delta d=v_{n}\times \Delta t$$

The new contour obtained is introduced in the subsequent iteration step. After a certain number of iterations, the final contour of the microgroove is obtained.

The absolute value of the difference between the taper value *θ* of the groove sidewall and 90° is used as the objective function *λ*(*x*), which is the arbitration index, while the cathode structure with the smallest objective function *λ*(*x*) is optimal.

An array microgroove structure with a depth of 0.52 mm, a width of 0.55 mm, and a sidewall taper of less than 2° was taken as the machining object to conduct the ECB process simulation analysis. *L*_1_ is the front tooth width of the wedge-shaped tooth tool, and *L*_2_ is the rear tooth width of the wedge-shaped tooth tool, which is fixed at 0.3 mm based on previous research. The front tooth width is changed to adjust the electric field distribution in the processing area to realize groove sidewall taper control. Therefore, the front tooth width is selected as the structural parameter of the cathode tool in this optimization.

### 3.2. Determining the Optimal Range of the Structural Parameter

Before optimizing the structural parameter, it is necessary to determine its range. Because the rear tooth width is determined to be 0.3 mm, the forming process simulations of the two-parameter values, namely front tooth widths of 0.3 mm and 0.1 mm, are carried out. The simulation parameters are shown in Table 2.

The simulation result for a conventional tool structure with equal front and rear tooth widths of 0.3 mm is shown in Figure 5. The anode edge point data at the bottom of the grooves are exported to the ORIGIN data processing software to fit the data points to generate the groove topography.

The groove width value of the cathode tool bottom and the horizontal height at the last moment of processing is recorded as *S*_1_, and the groove width value corresponding to the plane 0.1 mm below the workpiece machining plane is recorded as *S*_2_. The height difference between the two groove width lines is recorded as *T*. This method can effectively eliminate the influence of the groove sidewall upper and lower fillet radii on the taper measurement results. The measurement method is shown in Figure 5, and the sidewall taper value *θ* is calculated from Equation (11).
*θ* = arctan [(*S*_2_ − *S*_1_)/2*T*](11)

The measured sidewall taper of the cathode simulation electrolytic machining grooves with equal front and rear tooth widths is 6.22°. This sidewall taper exceeds the technical specifications and must be improved.

The simulation results of the proposed variable width tool structure, with a front tooth width of 0.1 mm and a rear tooth width of 0.3 mm, are shown in Figure 6. A noticeable reverse taper of −4.311° has appeared on the microgroove sidewall. The simulation results show that the sidewall taper of the grooves can be significantly changed by changing the front tooth width and reflecting the effectiveness of this method in regulating the sidewall taper. The simulation results with a front tooth width of 0.3 mm and 0.1 mm show that the sidewall taper value of the groove changes from a positive to a negative value. Because this is a single peak optimization problem, the optimal structural parameter value must be between 0.1 mm and 0.3 mm.

### 3.3. Optimal Cathode Tool Structural Parameter Value

The golden section method is an optimization algorithm based on the principles of symmetry and isometric contraction. The algorithm is simple and effective, can iterate and converge quickly and accurately, and is especially suitable for optimization problems with only one dependent variable. Therefore, this paper uses the golden section method to solve the cathode tool parameter optimization problem in micro multi-groove ECB. The initial left boundary is *a* = 0.1, the right boundary is *b* = 0.3, the iteration precision is set to *ε* = 0.05, and the number of iterations *m* is calculated from Equation (12), so six iterations are required.
(12)$$m\geq \frac{ln\left[\varepsilon /\left(b-a\right)\right]}{ln0.618}$$

The specific calculation steps are as follows:Step 1Define the initial interval as [*a*, *b*] and the iteration accuracy as *ε*.Step 2Take two points *a*_1_ and *a*_2_ in [*a*, *b*], where
(13)$$a_{1}=a+0.382\left(b-a\right)$$
(14)$$a_{2}=a+0.618\left(b-a\right)$$

Step 3Calculate *f*(*a*_1_) and *f*(*a*_2_).Step 4Compare the values of *f*(*a*_1_) and *f*(*a*_2_). If *f*(*a_1_*) *< f*(*a*_2_), eliminate the interval (*a*_2_, *b*], perform the replacements *b = a*_2_, *a*_2_
*= a*_1_, *f*(*a*_1_) *= f*(*a*_2_), and use Equation (13) to find a new point *a_1_*. If *f*(*a*_1_) *≥ f*(*a*_2_), eliminate the interval [*a*, *a*_1_), perform the replacements *a = a*_1_, *a*_1_
*= a*_2_, *f*(*a*_1_) *= f*(*a*_2_), and use Equation (14) to find a new point *a*_2_.Step 5After six iterations, stop the iteration when (*b* − *a*) ≤ *ε*, and use Equation (15) to obtain the optimal solution.


(15)
$$a^{*}=\frac{\left(a+b\right)}{2}$$


The details of the iterative process are provided in Table 3.

Selecting the boundary values *a* = 0.187 and *a*_2_ = 0.194 from the last iteration and substituting them into Equation (15), *a** is found to be 0.191. The final profile shown in Figure 7 is obtained after using this value to carry out an electric field simulation for the front tooth width. The measured sidewall taper is 0.286°, which meets the technical requirements. Compared with the traditional equal front and rear tooth width cathode, the sidewall taper is significantly reduced, verifying the rationality of the proposed method.

## 4. Experiment

The forming process simulation and cathode structure optimization results were verified by conducting micro multi-groove ECB experiments on a self-developed ECM machine tool. A 20% mass fraction NaNO_3_ solution is used as the electrolyte. The electrolyte inlet pressure is 0.8 MPa, and the electrolyte temperature is maintained at 25 °C. The remaining experimental parameters, consistent with the simulation parameters, are listed in Table 2.

The workpiece and cathode materials are both SUS304 stainless steel. The workpiece is 25 mm long, 20 mm wide, and 5 mm high. The three-dimensional model is shown in Figure 8. The wedge-shaped tooth sidewalls are electrophoretically plated with a 30-μm-thick insulating layer to prevent stray corrosion on the sidewall. The front face of the front guide block in the cathode tool assembly is inclined at an angle to the fixture flow channel to guide the electrolyte into the processing area. In addition, the guide block increases the electrolyte flow velocity and enhances its ability to wash away the ECM products.

A pointed conical guide structure is designed at the corner where the front guide block and the cathode tool meet to improve the uniformity of the electrolyte entering the machining gap channel. The diversion section at the bottom of the wedge block adopts a pointed diversion structure, and the flow channel at the bottom of the cathode edge is convergent, which enables the electrolyte to accelerate further after entering the flow channel, thus improving the flow field in the processing area, and reducing the probability of sparking. The flow field distribution between the flow channels is more uniform, and it helps to improve the processing accuracy and stability of the micro multi-grooves.

The cathode guide block is made of insulating material to shield the electric field corrosion so that the processing of a groove is prevented from affecting the adjacent grooves. Epoxy resin is used as the insulation layer for the cathode edge plating. The liquid epoxy resin is closely attached to the side wall of the cathode edge by electroplating so as to avoid the stray corrosion of the side wall of the cathode edge on the tank during the electrolytic broaching process. The three-dimensional structure of the cathode assembly and an exploded view are shown in Figure 9.

The fixture assembly shown in Figure 10 was designed considering the uniformity of the electrolyte flow field and the correctness of the positioning and clamping. The fixture is made of an epoxy resin material with good rigidity and insulating properties. The workpiece is fixed on the anode with conduction through the platen, and there are long guide sections on the front deflector to stabilize the flow field in the machining gap.

The processing process of the ECB micro multi-grooves is shown in Figure 11. At the initial stage of ECB, the cathode tool is located at a certain distance from the workpiece. The electrolyte flows into the flow channel inside the fixture at a high velocity from the inlet and is separated into multiple streams by the guide block to enter the machining gap of each microgroove. After applying the processing voltage, the cathode tool feeds horizontally to the workpiece at the feed rate of 120 mm/min. When the wedge-shaped tooth tool cuts into the workpiece, the anode material of the workpiece is eroded and dissolved, and all the micro-grooves will be processed at the same time.

The ECB specimens for the conventional and optimized cathode tool structures are shown in Figure 12 and Figure 13, respectively. A laser confocal microscope (VR-5000, Keyence, Osaka, Japan) is used to measure the geometric profiles of the micro multi-grooves. Each microgroove’s sidewall taper, depth, and width values are measured three times, and the average value is taken as the final.

It can be seen from the measurement data that the front and rear wedge-shaped tooth widths of the conventional cathode are equal. The average width is 563 μm, the average depth is 545 μm, and the average sidewall taper angle is 7.254°, which clearly exceeds the technical requirements. For the cathode tool with a variable tooth width, the front tooth width is 0.191 mm, the rear tooth width is 0.3 mm, and the sidewall taper is only 0.268°, meeting the technical requirements. The average width is 579 μm, and the average depth is 528 μm.

The surface roughness of the groove bottoms of the conventional and optimized cathode structures was detected using a Taylor Hobson surface roughness instrument (Lester, UK), and the measurement results of each groove were added to compute an average value. The average surface roughness of the conventional and optimized ECB tool structures are Ra 0.516 μm and Ra 0.512 μm, respectively. Therefore, it can be concluded that the experimental results confirm the correctness of the simulations, thereby demonstrating the effectiveness of the structural optimization.

## 5. Conclusions

A new cathode tool structure with a variable tooth width is proposed to improve the micro multi-groove ECB profile accuracy. Based on the simulation results of the ECB anode forming process, the optimal front tooth width is obtained using the golden section optimization method, which is 0.191mm, and the rear tooth width, which is set to 0.3 mm. The simulation results show that the proposed variable tooth width cathode tool significantly reduces the sidewall taper of the microgroove. Next, comparative tests of conventional and optimized cathode structures were carried out. At an ECB feed rate of 120 mm/min, microgrooves with an average width of 579 μm and depth of 528 μm were processed. The optimized variable tooth width tool reduced the sidewall taper of the grooves from 7.254° to 0.268°. The average surface roughness of the conventional and optimized ECB tool structures are basically the same, which are Ra 0.516 μm and Ra 0.512 μm, respectively. The experimental results verify the effectiveness of the simulations and cathode structure optimization.

## Figures and Tables

**Figure 1 micromachines-13-02062-f001:**
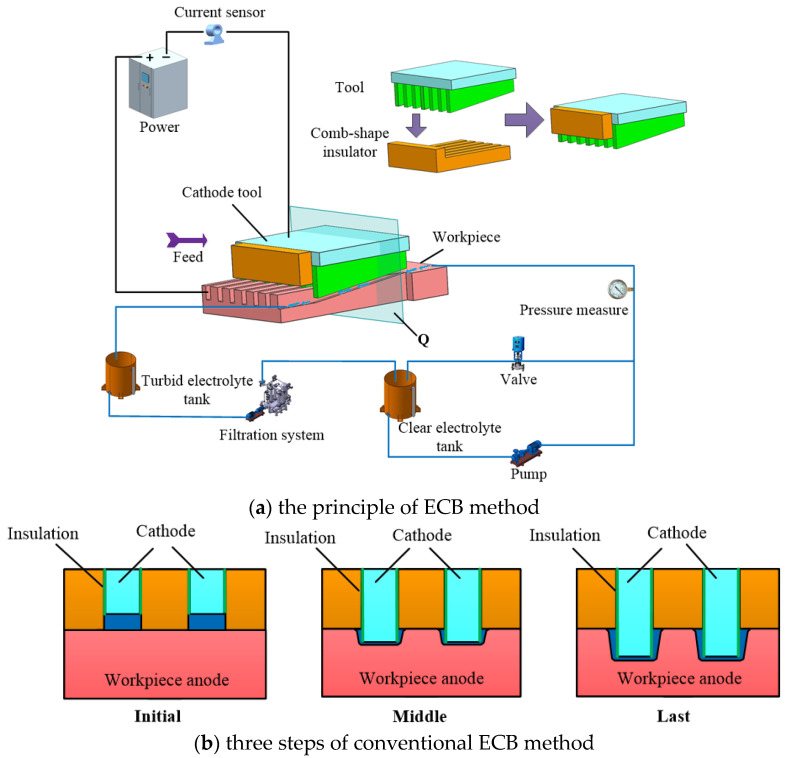
Schematic diagram of conventional ECB method for micro multi-grooves.

**Figure 2 micromachines-13-02062-f002:**
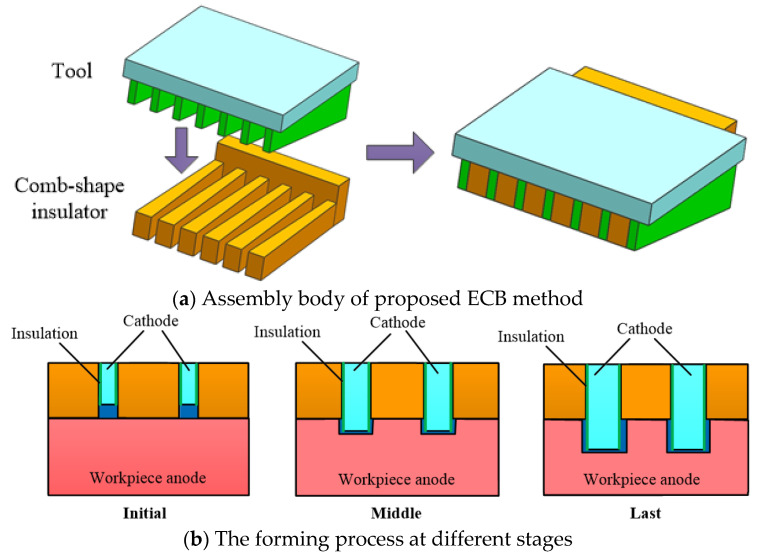
Schematic diagram of proposed ECB method for micro multi-grooves.

**Figure 3 micromachines-13-02062-f003:**
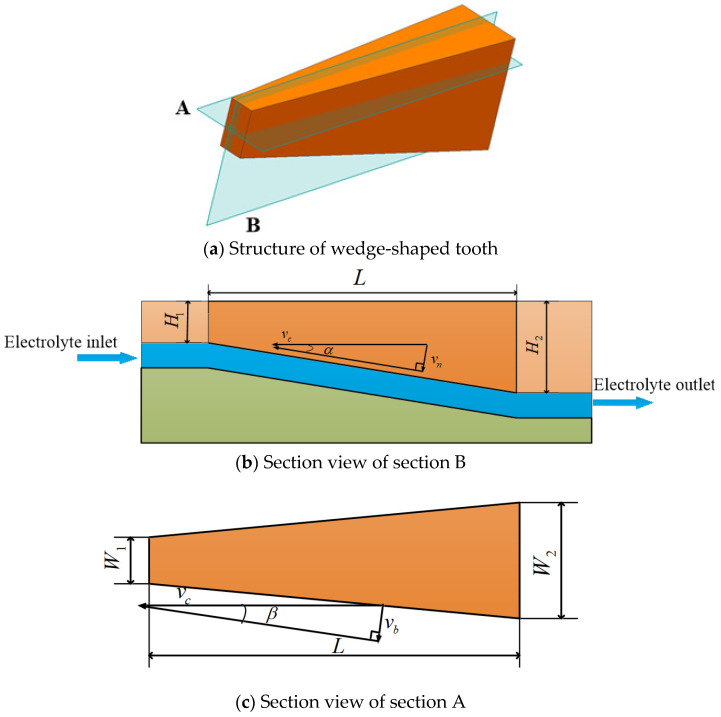
Schematic diagram of wedge-shaped tooth structure.

**Figure 4 micromachines-13-02062-f004:**
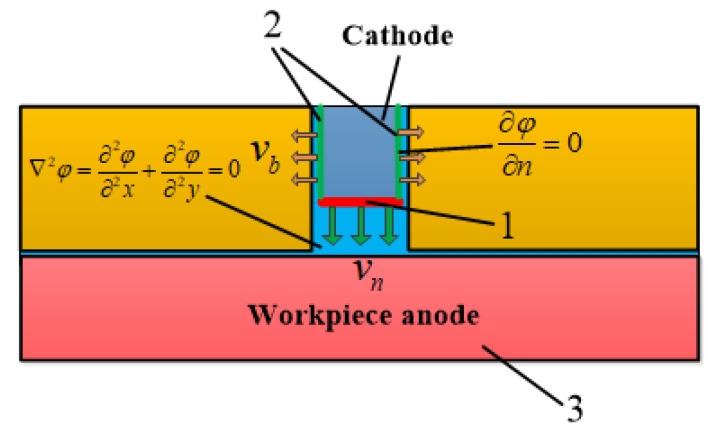
ECB dynamic forming simulation model.

**Figure 5 micromachines-13-02062-f005:**
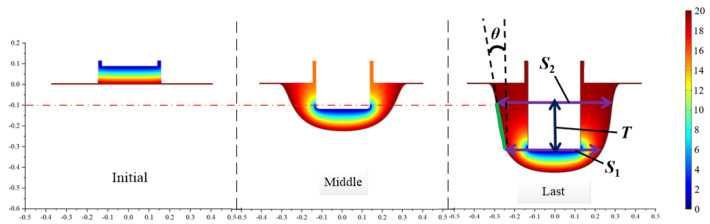
Simulation results for a conventional tool structure.

**Figure 6 micromachines-13-02062-f006:**
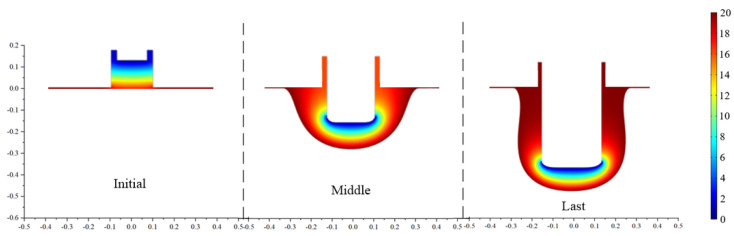
Simulation results of proposed tool structure with a 0.1 mm front tooth width.

**Figure 7 micromachines-13-02062-f007:**
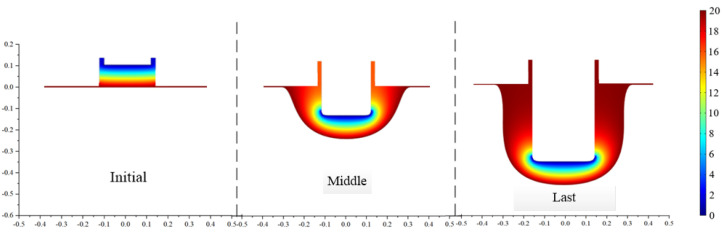
Simulation results of proposed tool structure with optimized structural parameter.

**Figure 8 micromachines-13-02062-f008:**
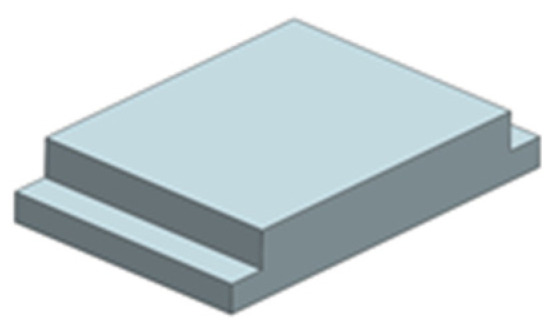
Three-dimensional model of the workpiece.

**Figure 9 micromachines-13-02062-f009:**
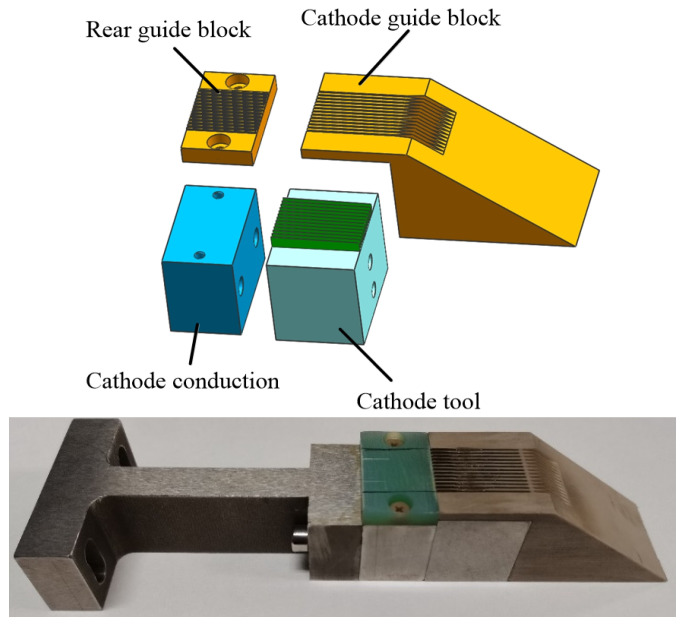
ECB cathode tool model assembly.

**Figure 10 micromachines-13-02062-f010:**
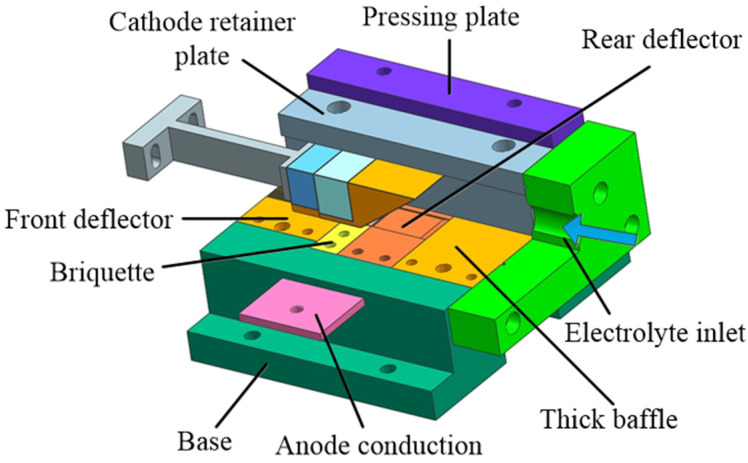
Fixture assembly model.

**Figure 11 micromachines-13-02062-f011:**
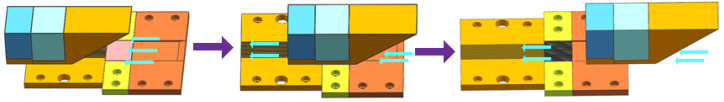
Schematic diagram of ECB process.

**Figure 12 micromachines-13-02062-f012:**
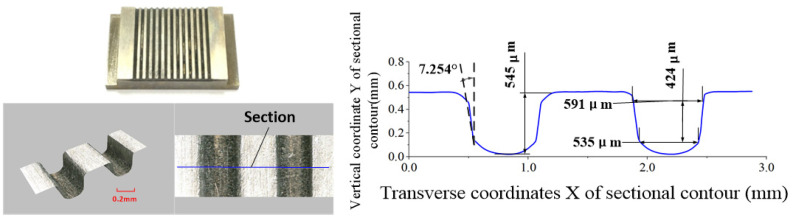
Front and rear edge equal-width cathode machining group groove specimens.

**Figure 13 micromachines-13-02062-f013:**
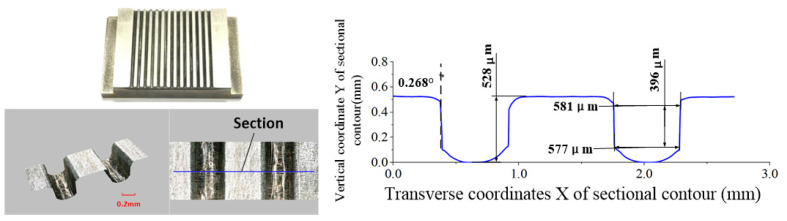
Specimen of cathode machining group grooves with 0.191 mm front edge widths.

**Table 1 micromachines-13-02062-t001:** ECB cathode tool structural parameters.

Structural Parameters	Value
*H* _1_	3 mm
*H* _2_	3.5 mm
*L*	20 mm
*W* _1_	0.3 mm
*W* _2_	0.3 mm

**Table 2 micromachines-13-02062-t002:** Simulation parameters.

Parameter Type	Value
Initial clearance (mm)	0.1
Processing voltage (V)	20
Electrolyte	20 wt% NaNO_3_
Electrolyte temperature (°C)	25
Feed rate (mm/min)	120
Front tooth width *W*_1_ (mm)	0.1, 0.3
Rear tooth width *W*_2_ (mm)	0.3

**Table 3 micromachines-13-02062-t003:** Golden section method iteration process.

Iteration Number	*a*	Point 0.382		Point 0.618		*b*	*ε*
		*a* _1_	*f*(*a*_1_)	*a* _2_	*f*(*a*_2_)		
	0.1	0.176	1.813	0.224	3.865	0.3	0.2
1	0.1	0.147	2.708	0.177	1.811	0.224	0.124
2	0.147	0.176	1.813	0.195	0.749	0.224	0.077
3	0.176	0.194	0.731	0.206	0.884	0.224	0.048
4	0.176	0.187	0.947	0.195	0.749	0.206	0.03
5	0.187	0.194	0.463	0.199	1.277	0.206	0.019
6	0.187	0.192	0.299	0.194	0.463	0.199	0.012

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
