# Peer review of "Study on the Control Method of Sidewall Taper in Electrolytic Broaching of Micro Multi-Grooves"

_micromachines, 2022, doi:10.3390/mi13122062_

Round 1
Reviewer 1 Report
This manuscript is aimed at the research of sidewall taper control technology in electrolytic broaching of micro multi-grooves. Overall presentation of the work is good, but there are few suggestions which I think are necessary to explain before publication.
1) Laser is an important processing technology of micro multi-grooves, and it is also widely used in industry. In the literature review, I suggest the authors to add the research progress of Laser micro groove technology.
2) The electrolyte flow field is an important factor to ensure the stability and accuracy of electrochemical machining. It should be introduced in detail how to realize the uniform distribution of flow field in each machining gap of micro groove.
3) The material of insulating layer on the wedge-shaped tooth sidewalls should be introduced in the manuscript.
Author Response
Reviewer #1
Reviewer’s comment 1
Laser is an important processing technology of micro multi-grooves, and it is also widely used in industry. In the literature review, I suggest the authors to add the research progress of Laser micro groove technology.
Response to comment 1
Thanks to the reviewer's reminder. We accept the reviewer's comments, add the relevant description in the introduction and add the literature citation.
1) Location: chapter 1 (Introduction), page 3, line 96 - 103.
Ultraviolet laser processing is also an important technology for machining micro grooves. The laser processing and manufacturing capability can reach the level of micrometer. Tang Yu et al.[27] developed a simple laser irradiation method to fabricate microscale titanium dioxide structure, and observed its morphology through scanning electron microscope and X-ray photoelectron spectroscopy. Through the analysis of high-speed photography, it is found that these micro grooves play an important role in enhancing water transmission performance, which provides inspiration for the application integration of such promising functional surfaces.
2) Location: References, page 14, line 465 - 466.
- Tang, Y.; Yang, X. L.; Wu, Y.; Zhu, D. Preparation of hierarchical Micro-Nano titanium dioxide structures via laser irradiation for enhancing water transport performance. Applied Surface Science, 2022, 586.
Reviewer’s comment 2
The electrolyte flow field is an important factor to ensure the stability and accuracy of electrochemical machining. It should be introduced in detail how to realize the uniform distribution of flow field in each machining gap of micro groove.
Response to comment 2
Thanks for the reviewer’s comment. We add the description of a pointed diversion structure in the chapter of experiment, as follows:
Location: chapter 4 (Experiment), page 11, line 327 - 332.
The diversion section at the bottom of wedge block adopts a pointed diversion structure, and the flow channel at the bottom of the cathode edge is convergent, which enables electrolyte to accelerate further after entering the flow channel, thus improving the flow field in the processing area and reducing the probability of sparking. The flow field distribution between the flow channels is more uniform, and it helps to improve the processing accuracy and stability of micro multi-grooves.
Reviewer’s comment 3
The material of insulating layer on the wedge-shaped tooth sidewalls should be introduced in the manuscript.
Response to comment 3
Thanks for the reviewer’s comment. We accept the reviewer's comments, as we all know, Epoxy resin coatings are widely used. They have excellent physical and electrical insulation properties, adhesion to metal materials and flexibility of plating process are not available for thermosetting plastics. Epoxy resin is extremely resistant to chemical corrosion, especially to alkali. It is immersed in electrolyte for a long time and its physical and chemical properties are not easy to change, which is very important for electrochemical processing. It has strong adhesion to metal materials and low viscosity. Impact resistance, not easy to deform and fall off under high water pressure; The epoxy resin has excellent heat resistance and electrical insulation, which is the most important advantage. So we add the description of the material of insulting layer in the chapter of experiment, as follows:
Location: chapter 4 (Experiment), page 11, line 334 - 338.
Epoxy resin is used as the insulation layer for cathode edge plating. The liquid epoxy resin is closely attached to the side wall of the cathode edge by electroplating, so as to avoid stray corrosion of the side wall of the cathode edge on the tank during the electrolytic broaching process.

Reviewer 2 Report
Overall review
1. The use of gold ratio 0.618 as the data source in line 271 lacks logic.
2. The experiment part is lack of detail.
3. The conclusion is not detailed enough.
Questions
1.Whether other data can replace 0.618 in the experiment?
2. Whether 335 lines of pictures can express fixture workflow through multiple pictures?
Author Response
Reviewer #2
Reviewer’s comment 1
The use of gold ratio 0.618 as the data source in line 271 lacks logic.
Response to comment 1
Thanks to the reviewer's comment. We accept the reviewer's comments and change the relevant description in the chapter 3(Simulation), as follows:
Location: chapter 3 (Simulation), page 9, line 279 - 282.
The golden section method is an optimization algorithm based on the principles of symmetry and isometric contraction. The algorithm is simple and effective, can iteratively converge quickly and accurately, and is especially suitable for optimization problems with only one dependent variable.
Reviewer’s comment 2
The experiment part is lack of detail.
Response to comment 2
Thanks for the reviewer’s comment. We accept the reviewer's comments and add the relevant description in the chapter 4 (Experiment), as follows:
1) Location: chapter 4 (Experiment), page 11, line 327 - 332.
The diversion section at the bottom of wedge block adopts a pointed diversion structure, and the flow channel at the bottom of the cathode edge is convergent, which enables electrolyte to accelerate further after entering the flow channel, thus improving the flow field in the processing area and reducing the probability of sparking. The flow field distribution between the flow channels is more uniform, and it helps to improve the processing accuracy and stability of micro multi-grooves.
2) Location: chapter 4 (Experiment), page 11, line 334 - 338.
Epoxy resin is used as the insulation layer for cathode edge plating. The liquid epoxy resin is closely attached to the side wall of the cathode edge by electroplating, so as to avoid stray corrosion of the side wall of the cathode edge on the tank during the electrolytic broaching process.
3) Location: chapter 4 (Experiment), page 12, line 353 – 360.
The processing process of ECB micro multi-grooves is shown in Figure 11. At the initial stage of ECB, the cathode tool is located at a certain distance from the workpiece. The electrolyte flows into the flow channel inside the fixture at a high velocity from the inlet, and is separated into multiple streams by the guide block to enter the machining gap of each micro groove. After applying processing voltage, the cathode tool feeds horizontally to the workpiece at the feed rate of 120mm/min. When the wedge-shaped tooth tool cuts into the workpiece, the anode material of the workpiece is eroded and dissolved, all micro grooves will be processed at the same time.
Reviewer’s comment 3
The conclusion is not detailed enough.
Response to comment 3
Thanks for the reviewer’s comment. We accept the reviewer's comments and add the relevant description in the conclusion.
1) Location: chapter 5 (Conclusion), page 13, line 389 - 391.
Based on the simulation results of the ECB anode forming process, the optimal front tooth width is obtained using the golden section optimization method, which is 0.191mm, and the rear tooth width is set to 0.3 mm.
2) Location: chapter 5 (Conclusion), page 13, line 397 - 398.
The average surface roughnesses of the conventional and optimized ECB tool structures are basically the same, which are Ra 0.516 μm and Ra 0.512 μm, respectively.
Reviewer’s question 1
Whether other data can replace 0.618 in the experiment?
Response to question 1
Thanks to the reviewer's reminder. We accept the reviewer's comments, as we know, 0.618 is a special golden section number, which has been widely used in engineering buildings since ancient times. The exterior configuration of the Parthenon Temple in the ancient Greek city-states largely uses the golden section; The pyramid of Egypt has a side ratio of 0.618 high to the quadrilateral at the bottom. The Oriental Pearl in Shanghai is a world-renowned project with a high ball height to total height ratio of 0.618. There are of course many more examples like this. Therefore, 0.618 was chosen because of its wide engineering significance.
Reviewer’s question 2
Whether 335 lines of pictures can express fixture workflow through multiple pictures?
Response to question 2
Thanks to the reviewer's reminder. We accept the reviewer's comments and add the relevant description in the chapter 4 (Experiment), we have revised Figure 10, the Figure 11 is attached in the chapter 4 (Experiment), as follows:
Location: chapter 4 (Experiment), page 12, line 351-352; page 12, line 353 – 360; page 12, line 361 – 362.
The processing process of ECB micro multi-grooves is shown in Figure 11. At the initial stage of ECB, the cathode tool is located at a certain distance from the workpiece. The electrolyte flows into the flow channel inside the fixture at a high velocity from the inlet, and is separated into multiple streams by the guide block to enter the machining gap of each micro groove. After applying processing voltage, the cathode tool feeds horizontally to the workpiece at the feed rate of 120mm/min. When the wedge-shaped tooth tool cuts into the workpiece, the anode material of the workpiece is eroded and dissolved, all micro grooves will be processed at the same time.
Figure 10. Fixture assembly model.
